# Risk factors associated with congenital anomalies among newborns in southwestern Ethiopia: A case-control study

**Soressa Abebe** [1]*, **Girmai Gebru**[1], **Demisew Amenu**[2], **Zeleke Mekonnen**[3], **Lemessa Dube**[4]

**1** Department of Anatomy, School of Medicine, College of Health Sciences, Addis Ababa University, Addis Ababa, Ethiopia, **2** Department of Gynecology and Obstetrics, Institute of Health, Jimma University, Jimma, Oromia, Ethiopia, **3** School of Medical Laboratory Sciences, Institute of Health, Jimma University, Jimma, Oromia, Ethiopia, **4** Department of Epidemiology, Institute of Health, Jimma University, Jimma, Oromia, Ethiopia

* abebe.soressa@gmail.com

## Abstract

### Introduction

Human embryo is well protected in the uterus by the embryonic membrane, although teratogens may cause developmental disruptions after maternal exposure to them during early pregnancy. Most of the risk factors contributing to the development of congenital anomalies are uncertain; however, genetic factors, environmental factors and multifactorial inheritance are found to be risk factors. Regardless of their clinical importance, there are little/no studies conducted directly related to predisposing risk factors in southwestern Ethiopia.

### Objective

The study aimed to determine the associated risk factors with congenital anomalies among newborns in southwestern Ethiopia.

### Methods

Case—control study was conducted on newborns and their mothers in six purposively selected hospitals in southwestern Ethiopia from May 2016 to May 2018. Data was collected after evaluation of the neonates for the presence of congenital anomalies using the standard pretested checklist. The data was analyzed using SPSS version 25.0. P <0.01 was set as statistically significant.

### Results

Risk factors such as unidentified medicinal usage in the first three months of pregnancy (AOR = 3.435; 99% CI: 2.012–5.863), exposure to pesticide (AOR = 3.926; 99% CI: 1.266–12.176), passive smoking (AOR = 4.104; 99% CI: 1.892–8.901), surface water as sources of drinking (AOR = 2.073; 99% CI: 1.221–3.519), folic acid supplementation during the early

**Data Availability Statement:** All relevant data are within the manuscript and its supporting information files.

**Funding:** This study obtained fund from Addis Ababa University and Jimma University for data collection only. The funders had no role in the study design, data analysis and decision to publish for preparation of the manuscript. No additional external funding was received for this study.

**Competing interests:** The authors have declared that no competing interests exist.

pregnancy (AOR = 0.428; 99% CI: 0.247–0.740) were significantly associated with the congenital anomalies.

## Conclusions

In this study, risk factors such as passive smoking, exposure to pesticides, chemicals and use of surface water as a source of drinking during early pregnancy had a significant association with congenital anomalies. There is a need to continuously provide health information for the community on how to prevent and control predisposing risk factors.

## Introduction

Intrauterine development can be considered as normal development as well as abnormal development. Abnormal development occurs because of the interference of normal development from genetic disorders, environmental factors, and the combination of both genetic and environmental factors during the critical period of embryogenesis. This leads to abnormal cytogenesis, histogenesis and morphogenesis with which the neonate born with a defect known as a congenital anomaly (CA) [1,2].

CA begins to emerge as one of the major childhood health problems as well as a cause of infant mortality and morbidity throughout the world especially in developing countries [3]. In most cases, infants with malformation do not survive, more than 70% die in the first month of life. Moreover, treatment and healing of children with congenital malformations are costly and complete recovery may be impossible [4–8].

However, most of the risk factors of CAs are uncertain (40–60%) genetic factors, environmental factors, and multifactorial inheritance are among risk factors that lead to abnormal prenatal development [4]. Maternal chromosomal abnormalities which can be defined as numerical and structural abnormalities of the chromosomes cause a genetic disorder. Besides, a single gene mutation also contributes to genetic related birth defects [2].

CAs are a major cause of prenatal and neonatal deaths worldwide [1]. The prevalence rates of all genetic birth defects combined range from a high of 82/1,000 live births in low–income regions to a low of 39.7/1,000 live births in high–income regions [1,9].

The human embryo is well protected in the uterus by the extra-embryonic membranes, although teratogens/ environmental factors may cause developmental disruptions after maternal exposure to them in a specific period of embryogenesis during the critical period in early pregnancy [4]. The organs or parts of an embryo are most sensitive to environmental factors during periods of rapid differentiation leading to abnormal development or malformation of those organs [10,11].

Environmental factors that are considered as being potential risk factors in causing congenital malformation include maternal infection, maternal age and maternal drug intake during the critical period of embryogenesis and substances such as, caffeine, nicotine, commonly used medicines, maternal nutritional and health status, maternal exposure to hazardous waste and maternal alcohol intake during early pregnancy [1,2]. Moreover, parental race, parental socio-economic status, hyperthermia during early pregnancy and maternal diabetes as well as obesity are also considered to be associated risk factors in causing developmental malformations [2].

About 10% of CAs are caused due to environmental factors. The risk identification for exposure to such factors is of paramount importance as some forms of CAs can be prevented to a large extent if identified and appropriate caution is taken [1].

Chemical substances such as mercury, lead and arsenic are known to lead to the development of congenital abnormalities [10]. Mercury, which is found in some types of fish, has been linked with the development of neurological problems resembling cerebral palsy as well as mental retardation. Lead has been associated with fetal growth restriction and neurological disorders [2].

Other environmental factors such as radiation also contribute to the formation of abnormal development. For example, X-rays can cause problems with fetal development, such as spina bifida, cleft palate, blindness, abnormalities of the arms and legs, or microcephaly. But the type of malformation or abnormality that develops depends on the dose of X-ray radiation that the pregnant woman is exposed to during early pregnancy [1,4].

Multifactorial inheritances, referring to the additive effects of many genetic and environmental factors, are responsible for several developmental disorders resulting in congenital malformations [11,12]. Multifactorial inheritances linked to the causation of CAs in humans include gene-gene and gene-environment interactions and have been demonstrated in mouse models of neural tube defects [11].

CAs are the fifth leading cause of years of potential life loss and a major cause of morbidity and mortality throughout children of the world. Regardless of their clinical importance, there are little/no studies conducted directly related to predisposing risk factors including genetic diseases/disorders that could demonstrate the incidence and prevalence level of these conditions and related risk factors in Ethiopia. Hence, to realize the organization of community genetic services program in primary health care in Ethiopia, conducting hospital-based studies that may give a clear picture of predisposing risk factors associated with a CA is needed to be evaluated. Furthermore, there is a paucity of information regarding the predisposing risk factors associated with CAs specifically in southwestern Ethiopia. Thus, the present study was carried out to investigate risk factors associated with CAs in southwestern Ethiopia.

## Subjects and methods

### Study area and period

The study was conducted in southwestern Ethiopia from May 2016 to May 2018 for two years during which there were 35,080 deliveries. Southwestern Ethiopia mainly belongs to the Oromia regional state–the largest region with the largest population sizein Ethiopia. Most of the population in the southwestern region are Muslims, Orthodox, Protestants, and some others.

### Design of the study and sample size

The case-control study design was used to investigate the exposure status of associated risk factors for CAs in case and control newborns (alive and stillbirth) in six out of seven hospitals in southwestern Ethiopia. Cases were births (fresh still or alive) with minor and major CAs, while the controls were births (fresh still or alive) without any CAs. Two hundred fifty-one cases and 887 controls were used with a total sample size of 1138. The sample size was calculated using a case-control ratio of 1:4. The proportion difference approach with the desired confidence level of 99%, 80% power, and a 1% significant level, assuming that the Odds ratio expected to be 2 [13]. Whenever a case occurred, four controls were randomly selected from the existing newborns without CAs at the maternity ward immediately after the occurrence of the case/cases.

### Source and study population

All new births that were delivered and their corresponding mothers who had given birth in the sampled hospitals in southwestern Ethiopia during the study period were the source

population for the study. Source populations for cases were all births (still or live) with CAs and mothers who had a neonate with CAs in the selected hospitals during the study period. Source populations for controls were all births (still and live births) without any CAs and their corresponding mothers in the selected hospitals during the study period. All the still and alive born neonates in the selected hospitals during the study period were included in the study.

## Inclusion criteria

### Cases

- Births (live birth or fresh stillbirth) either singletons or multiple at terms of gestational age with any CAs or disorders.

### Controls

- Births (live birth or fresh stillbirth) either singletons or multiple at terms of gestational age without any congenital disorders, but in the same geographical location with the cases.

## Exclusion criteria

### Cases and controls

A mother that came from a different region was excluded from both case and control.

### Sampling techniques

There were seven functional hospitals in the study area. We have purposively included all except one in the study after reviewing the hospital record for CAs and capacity to assess the defects among the newborns. These are Jimma University specialized teaching hospital, Shanan Gibe hospital, Agaro hospital, Limmu Genet hospital, Nekemte hospital, and Mettu Karl hospital. Then cases and controls were selected consecutively.

### Data collection and measurement

Newborns, either fresh stillbirth or live birth in the selected hospitals during the study period were evaluated for the presence of any CAs by trained health professionals. A standard pretested checklist was used to assess every live and fresh stillbirth at selected hospitals. The assessment of CAs was conducted by trained residents, general practitioners, and midwives at obstetrics and gynecology delivery wards in study hospitals. The data were collected from the mothers of cases and controls via face—to—face interviews using a structured questionnaire containing variables regarding socio-demographic, maternal characters, neonate characters, and other associated risk factors.

## Study variables

### Dependent variables

Congenital anomalies (yes/no)

Types of congenital anomalies

 **Independent variables.** Maternal age, socioeconomic status, educational background, maternal health status (medical disorder such diabetes mellitus), maternal exposure to drugs,

maternal exposure to infection, maternal exposure to pesticides, medications, alcohol, tobacco, khat and waste disposal areas and sources of drinking water, mode of deliveries, gestational age, sex and birth weight of the neonate were considered as associated risk factors for the CAs (malformation) and were the focus of the study.

**Data management and analysis.** The collected data were checked regularly on a daily basis, whenever the case occurred, by the data collectors and further superintended by the principal investigator and supervisors for its completeness and correctness. The collected data were cleaned, coded and entered into Epi Data manager computer software and transferred to SPSS Version 25.0 for analysis. The outcome of fetal birth disorder was determined to intermesh with the mode of deliveries. CAs were classified according to ICD-10 (International Classification of Diseases– 10). Total prevalence was calculated by dividing the numerator (registered cases of CA) by the relevant denominator (total live and stillbirths) for the same period of time at the same place. Overall data were calculated using frequency, cross-tabulation, binary and multiple logistic regressions. Exposure variables with $P$-value $\leq 0.2$ on bivariate analysis were entered into a multivariable logistic regression model to evaluate the association between the exposures and CAs. The COR and AOR with their 99% confidence intervals and the $P$-value as well as the results of the findings were presented in the form of text and table. The 99% CI not including the null value was used to declare statistical significance.

**Quality assurance.** The questionnaire was prepared in English then translated into Amharic and Afaan Oromo and translated back into English by another person to check its semantic equivalence. The questionnaire was pre-tested on 5% of the sample size in the non-selected health center district. Training was given to data collectors and evaluators. The completed questionnaire was checked for consistency and completeness by the principal investigator and supervisors.

**Ethical clearance.** Ethical approval was obtained from the Addis Ababa University, College of Health Sciences, Institutional Review Board, meeting Ref. No. 005/16, dated May 2016. Supportive letters were written and submitted to all study hospitals' administrators. Then data collections were started after permissions were obtained from the medical directors of each hospital. The study procedure and its aim were disclosed to participants. All the study participants were informed that participation is voluntary and can be withdrawn at any time. Not being volunteers cannot affect the service that they did get from the hospital. Written and signed consent was obtained from all study participants before administering the questionnaires and face—to—face interviews. The participants did not write their names on the questionnaires for confidentiality and no identifying information was recorded by participants. The data were coded, stored in a safe and secure location. The data were never exposed to anyone and only used for the study purpose to maintain confidentiality.

## Results

In the present study, the sample consisted of 251 newborns with CAs and 887 newborns without CAs. The case to control ratio used was 1:4. Twenty-four types of CAs were identified with total anomalies of 290 during the study period. Of 887 controls, 628(55.2%) were male and 510 (44.8%) were female newborns. 128 (51.0%) and 123 (49.0%) of the cases were male and female, respectively. About 212 (84.5%) of the CAs identified were single/isolated, whereas 39 (15.5%) were multiple—two or more anomalies on a single case involving two or more organ systems. The overall incidence rate of CAs in southwestern Ethiopia was 71.6 per 10,000 births. The overall frequency of CAs by organ system is shown in Table 1.

**Table 1. Frequency of congenital anomalies by organ/organ system among study subjects in southwestern Ethiopia.**

| Types of congenital anomalies | Frequency | % |
|---|---|---|
| Neural tube defects | 177 | 15.55 |
| Musculoskeletal defects | 46 | 4.04 |
| Gastrointestinal defects | 16 | 1.41 |
| Urogenital defects | 6+1* = 7 | 0.62 |
| CHD | 1 | 0.09 |
| Genetic disorder: Down syndrome, Achondroplasia | 4 | 0.35 |
| No congenital anomalies/controls | 887 | 77.94 |
| Total | 1138 | 100 |

CHD: Congenital heart defect;1*: Ambiguous genitalia.

## Socio-demographic characteristics of the study subjects

The mean age for case and control mothers was 26 years old. The maternal age of the cases was ranging from 17 to 42 years. Similarly, the maternal age of controls was ranging from 15 to 40 years old. Likewise, the paternal age range for cases and controls was 20 to 61 years old and 19 to 66 years old, respectively.

About 26.0% and 21.0% of mothers of the cases and controls were below 20 years old, respectively. Whereas, 32.4% and 37.5% of mothers of the cases and controls were in the age group of 21 to 24 years old, respectively. 38.0% and 37.4% of the mothers of the cases and controls were in the age group of 26 to 35 years old, respectively. Lastly, about 3.6% of both mothers of the cases and controls were ≥36 years old (Table 2).

Among the mothers of cases and controls, about 60.6% had formal education while 39.4% of the mothers were illiterate. 5% had primary education, 31.5% had junior level, 38.1% had attained high school level and 25.5% had joined higher education. In terms of religious inference, 61.6%, 23.8%, and 14.6% were Muslims, Orthodox, and Protestants, respectively.

71.8%, 6.7%, and 1.4% had a monthly income of twenty-seven to one hundred thirty-five, one hundred sixty-two to two hundred seventy and two hundred ninety seven and above USD, respectively. Among the mothers of both cases and controls, 59.9% were housewives, 11.8% were farmers, 18.9% were governmental employees, 7.6 were merchants and 1.7% were unemployed. The socio-demographic characteristics of the study participants of cases and controls are shown in Table 2.

About 7.2% of the mothers of the cases and 2.9% of mothers of controls had no antenatal care visit. Regarding the birth order of a case, 37.2%, 21.1%, 14.2%, and 27.0% were the first, the second, the third, the fourth, and above (4+) babies to their family, respectively. For those of the control group, about 43.6%, 24.8%, 13.0%, and 18.5% were the first, second, third, fourth, and above (4+) babies to their family, respectively.

Regarding gestational age, about 47.6% of cases and 19.9% of the controls were classified as preterm (<37 weeks). The differences between the cases and the controls were statistically significant with an Odds ratio of 2.461; 99% CI: 1.806–3.299, P-value < 0.001, revealing that cases are more likely to have a premature birth. Hence, premature births were associated with the presence of CA. On the contrary, 40.8% of the cases and 58.1% of the controls attained their full-term gestational age (Table 3).

As shown in Table 3, 39.8% and 8.2% of the cases and controls were born with low birth weight (<2500 g), respectively. About 61.6% and 5.2% of the cases and controls were sillbirth, respectively. There was a significant difference between the cases and the controls revealing

**Table 2. Socio-demographic characteristics of the study subjects with or without CAs in southwestern Ethiopia.**

| Characteristics | | Cases (n = 251) | | Controls (n = 887) | | Total (n = 1138) | |
|---|---|---|---|---|---|---|---|
| | | Frequency | % | Frequency | % | Frequency | % |
| Age of the mother | < = 20 | 65 | 26.0 | 189 | 21.4 | 254 | 22.4 |
| | 21–25 | 81 | 32.4 | 331 | 37.5 | 412 | 36.4 |
| | 26–35 | 95 | 38.0 | 330 | 37.4 | 425 | 37.5 |
| | > = 36 | 9 | 3.6 | 32 | 3.6 | 41 | 3.6 |
| Maternal education | Illiterate | 120 | 47.8 | 327 | 37.0 | 447 | 39.4 |
| | Literate | 131 | 52.2 | 556 | 63.0 | 687 | 60.6 |
| Maternal educational level | 1–4 (primary) | 13 | 9.9 | 21 | 3.8 | 34 | 5.0 |
| | 5–8(junior) | 46 | 35.1 | 169 | 30.6 | 215 | 31.5 |
| | 9–12 (high school) | 49 | 37.4 | 211 | 38.2 | 260 | 38.1 |
| | 13 and above (Higher education) | 23 | 17.6 | 151 | 27.4 | 174 | 25.5 |
| Religion | Muslim | 164 | 65.3 | 535 | 60.6 | 699 | 61.6 |
| | Orthodox | 48 | 19.1 | 222 | 25.1 | 270 | 23.8 |
| | Protestant | 39 | 15.5 | 126 | 14.3 | 165 | 14.6 |
| Average monthly income in USD | <27 | 37 | 26.4 | 89 | 18.3 | 126 | 20.1 |
| | 27–135 | 95 | 67.9 | 355 | 72.9 | 450 | 71.8 |
| | 162–270 | 6 | 4.3 | 36 | 7.4 | 42 | 6.7 |
| | 297 and above | 2 | 1.4 | 7 | 1.4 | 9 | 1.4 |
| Maternal occupation | Housewife | 151 | 64.8 | 503 | 58.6 | 654 | 59.9 |
| | Farmer | 34 | 14.6 | 95 | 11.1 | 129 | 11.8 |
| | Employee | 29 | 12.4 | 177 | 20.6 | 206 | 18.9 |
| | Merchant | 11 | 4.7 | 72 | 8.4 | 83 | 7.6 |
| | Unemployed | 8 | 3.4 | 11 | 1.3 | 19 | 1.7 |
| Fathers age | < = 25 | 55 | 22.6 | 236 | 29.7 | 291 | 28.0 |
| | 26–34 | 104 | 42.8 | 327 | 41.1 | 431 | 41.5 |
| | > = 35 | 84 | 34.6 | 232 | 29.2 | 316 | 30.4 |

USD: United State Dollar.

that the presence of CAs could affect the neonatal outcome. Hence, more stillbirths occurred in cases than controls.

In the present study, about 12.4% and 9.2% of mothers of the cases and controls had a history of abortion. However, the difference between mothers of the cases and controls were not statistically significant. 9.3% and 7.9% of mothers of the cases and controls had a history of stillbirth, respectively. About 3.0% and 0.8% of the cases and control mothers had a birth history of CAs, respectively indicating that there were no statistically significant differences between the cases and the control mothers (COR = 3.607; 99% CI: 0.899–14. 471; $P$-Value = 0.017) (Table 4).

About 3.8% of mothers of the cases and 1.7% of controls had a history of congenital malformation in their family, revealing that there were statistically significant differences between the case mother and control mothers (COR = 2.685; 99% CI: 0.949–7.595; P-value = 0.014).

## Associated risk factors with congenital anomalies

In the present study, there were maternal illnesses in 113 (45.0%) mothers of the cases and 373 (42.2%) of controls. 15 (6.0%) mothers of the cases and 48 (5.4%) of the controls had a history of taking alcohol during early pregnancy/first trimester of pregnancy. 3 (1.2%) mothers of the cases and 2(0.2%) controls mothers had a history of smoking cigarettes.

**Table 3. Reproductive history of mothers who gave birth to a child with or without CAs in southwestern Ethiopia.**

| Characteristics | | Case (n = 251) | | Control (n = 887) | | Total (1138) | |
|---|---|---|---|---|---|---|---|
| | | Frequency | % | Frequency | % | Frequency | % |
| Antenatal care follow up | | | | | | | |
| No antenatal care | | 18 | 7.2 | 26 | 2.9 | 44 | 3.9 |
| 1 to 3 visits | | 138 | 55.5 | 311 | 35.2 | 449 | 39.7 |
| Minimum of 4 visits | | 94 | 37.6 | 546 | 62.0 | 640 | 56.5 |
| Parity | 0 & 1 | 13 | 54.5 | 537 | 61.7 | 671 | 60.1 |
| | 2 & 3 | 68 | 27.6 | 217 | 24.9 | 285 | 25.5 |
| | > = 4 | 44 | 17.9 | 117 | 13.4 | 161 | 14.4 |
| Gravid | 0 & 1 | 90 | 36.4 | 377 | 43.4 | 467 | 41.8 |
| | 2 & 3 | 83 | 33.6 | 321 | 36.9 | 404 | 36.2 |
| | > = 4 | 44 | 30.0 | 171 | 19.7 | 245 | 22.0 |
| Onset of labor | Spontaneous | 197 | 79.1 | 799 | 91.6 | 996 | 88.8 |
| | Induced | 52 | 20.9 | 73 | 8.4 | 125 | 11.2 |
| Mode of delivery | Vaginal | 208 | 83.2 | 629 | 71.5 | 837 | 74.1 |
| | Scissoral | 42 | 16.8 | 251 | 28.5 | 293 | 25.9 |
| Gestational age | Preterm | 119 | 47.6 | 176 | 19.9 | 295 | 26.0 |
| | Term | 102 | 40.8 | 514 | 58.1 | 616 | 54.3 |
| | Post-term | 29 | 11.6 | 195 | 22.0 | 224 | 19.7 |
| Sex of the newborn | Male | 128 | 51.0 | 500 | 56.6 | 628 | 55.4 |
| | Female | 123 | 49.0 | 383 | 43.4 | 506 | 44.6 |
| Birth weight in gram | <2500 | 100 | 39.8 | 72 | 8.2 | 172 | 15.2 |
| | > = 2500 | 151 | 60.2 | 811 | 91.8 | 962 | 84.8 |
| Birth outcome | Live birth | 96 | 38.4 | 836 | 94.8 | 932 | 82.3 |
| | Stillbirth | 154 | 61.6 | 46 | 5.2 | 200 | 17.7 |
| Types of birth outcome | Single | 236 | 94.8 | 836 | 96.3 | 1072 | 96.0 |
| | Twin | 13 | 5.2 | 32 | 3.7 | 45 | 4.0 |
| Birth order | 1st | 92 | 37.2 | 380 | 43.6 | 472 | 42.2 |
| | 2nd | 52 | 21.1 | 216 | 24.8 | 268 | 23.9 |
| | 3rd | 35 | 14.2 | 113 | 13.0 | 148 | 13.9 |
| | 4th and greater | 67 | 27.0 | 163 | 18.5 | 230 | 19.5 |

Passive smokers were observed in 28(11.2%) mothers of the cases and 28 (3.2%) of the controls. 5(2.0%) mother of the cases and 5(0.6%) of the controls had a history of exposure to

**Table 4. Obstetrics history of the study subjects in southwestern Ethiopia.**

| Variables | | Cases | | Controls | | OR (99% CI) | P-value |
|---|---|---|---|---|---|---|---|
| | | Number | % | Number | % | | |
| History of abortions | Yes | 29 | 12.4 | 81 | 9.2 | 1.467 (0.827–2.601 | 0.085 |
| | No | 205 | 87.6 | 799 | 90.8 | 1 | |
| History of stillbirths | Yes | 22 | 9.3 | 70 | 7.9 | 1.229 (0.649–2.330) | 0.405 |
| | No | 215 | 90.7 | 811 | 92.1 | 1 | |
| Previous history of a congenital anomaly | Yes | 7 | 3.0 | 7 | 0.8 | 3.607 (0.899–14.471) | 0.017 |
| | No | 230 | 97.0 | 877 | 99.2 | 1 | |
| History of congenital anomalies in the family | Yes | 9 | 3.8 | 15 | 1.7 | 2.685 (0.949–7.595) | 0.014 |
| | No | 225 | 96.2 | 861 | 98.3 | 1 | |

OR: Odds ratio; CI: Confidence interval.

radiation. Exposure to pesticides and the use of different antibiotics during their early pregnancy/ the first three months were observed in 13(5.2%) and 79 (53.4%) mothers of the cases, respectively. Whereas, 12(1.4%) and 261(46.9%) of the controls had exposure to pesticides and used different antibiotics during their early pregnancy, respectively.

57(22.7%) and 77(8.7%) of mothers of the cases and mothers of the controls had experiences of using unidentified medicine and drugs during the first three months of their pregnancy, respectively. Similarly, 47(19.0%) and 38(15.3%) mothers of the cases and 144(16.4%) and 172(19.5%) of the controls had a history of using drugs in the second and third trimester of their pregnancy, respectively (Table 5).

162(64.5%) mothers of the cases and 449(50.8%) of controls were not used folic acid supplementation during the index pregnancy, respectively. Likewise, diabetes mellitus

**Table 5. Bivariate analysis of environmental, family history, exposure to different chemicals, and maternal illness of the study subjects in southwestern Ethiopia.**

| Variables | Response | Cases (n = 251) | | Controls (n = 887) | | COR | 99% CI | | P-value |
|---|---|---|---|---|---|---|---|---|---|
| | | Number | % | Number | % | | Lower | Upper | |
| Folic acid use | Yes | 89 | 35.5 | 434 | 49.2 | 0.563 | 0.385 | 0.825 | 0.000 |
| | No | 162 | 64.5 | 449 | 50.8 | Ref 1 | | | |
| Drinking alcohol | Yes | 15 | 6.0 | 48 | 5.4 | 1.115 | 0.508 | 2.446 | 0.721 |
| | No | 234 | 94.0 | 833 | 94.6 | 1 | | | |
| Smoking cigarettes | Yes | 3 | 1.2 | 2 | 0.2 | 5.317 | 0.503 | 56.242 | 0.068 |
| | No | 248 | 98.8 | 870 | 99.8 | 1 | | | |
| Passive smoking | Yes | 28 | 11.2 | 28 | 3.2 | 3.852 | 1.884 | 7.875 | 0.000 |
| | No | 223 | 88.8 | 855 | 96.8 | 1 | | | |
| Exposure to X–rays | Yes | 5 | 2.0 | 5 | 0.6 | 3.586 | 0.696 | 8.482 | 0.045 |
| | No | 244 | 98.0 | 873 | 99.4 | 1 | | | |
| Exposure to pesticides | Yes | 13 | 5.2 | 12 | 1.4 | 4.012 | 1.406 | 11.446 | 0.001 |
| | No | 236 | 94.8 | 871 | 98.6 | 1 | | | |
| Diabetes mellitus | Yes | 4 | 1.6 | 1 | 0.1 | 14.341 | 0.800 | 256.981 | 0.017 |
| | No | 246 | 98.4 | 880 | 99.9 | 1 | | | |
| Maternal illness | Yes | 113 | 45.0 | 373 | 42.2 | 1.118 | 0.772 | 1.620 | 0.438 |
| | No | 138 | 55.0 | 510 | 57.8 | 1 | | | |
| Use of antibiotics | Yes | 79 | 53.4 | 261 | 46.9 | 1.303 | 0.808 | 2.100 | 0.153 |
| | No | 69 | 46.6 | 296 | 53.1 | 1 | | | |
| Have asthma | Yes | 4 | 1.6 | 15 | 1.7 | 0.648 | 0.127 | 3.309 | 0.493 |
| | No | 247 | 98.8 | 852 | 98.3 | 1 | | | |
| Drug use during the first three months | Yes | 57 | 22.7 | 77 | 8.7 | 3.091 | 1.884 | 5.070 | 0.000 |
| | No | 194 | 77.3 | 806 | 91.3 | 1 | | | |
| Drug use between 4th and 6th months | Yes | 47 | 19.0 | 144 | 16.4 | 1.197 | 0.741 | 1.932 | 0.334 |
| | No | 201 | 81.0 | 736 | 83.6 | 1 | | | |
| Drug use during the last three months | Yes | 38 | 15.3 | 172 | 19.5 | 0.747 | 0.451 | 1.237 | 0.136 |
| | No | 210 | 84.7 | 709 | 80.5 | 1 | | | |
| Drinking coffee during pregnancy | Yes | 218 | 92.4 | 780 | 88.3 | 1.353 | 0.716 | 2.554 | 0.221 |
| | No | 18 | 7.6 | 103 | 11.7 | 1 | | | |
| Use of khat during pregnancy | Yes | 23 | 9.2 | 54 | 6.1 | 1.556 | 0.797 | 3.040 | 0.089 |
| | No | 228 | 90.8 | | | 1 | | | |
| Hypertension disorder | Yes | 16 | 6.4 | 52 | 5.9 | 1.091 | 0.510 | 2.335 | 0.767 |
| | No | 234 | 93.6 | 829 | 94.1 | 1 | | | |

Reference = 1.

was observed in 4 (1.6%) mothers of cases and 1(0.1%) in controls. Besides, 16(6.4%) mothers of the cases and 52(5.9%) of the control had hypertension before and during pregnancy. Asthma was observed in 4 (1.6%) mothers of the cases and 15(1.7%) of the controls.

Selected variables were entered in to the COR analysis to identify crude risk estimates. Of these, smoking cigarettes during pregnancy (COR = 5.317; 99% CI: 0.503–56.242, *P*-value = 0.068), passive smoking (COR = 3.852; 99% CI: 1.884–7.875, *P*-value = <0.001), exposure to radiation (X–rays) in the early pregnancy (COR = 3.586; 99% CI: 0.696–8.482, P-value = 0.045), exposure to pesticides (COR = 4.012; 99% CI: 1.406–11.446, P–value <0.001), diabetic mellitus (COR = 14.341; 99% CI: 0.800–256.981, P- value = 0.017), use of unidentified medication and drugs in the first three months of pregnancy (COR = 3.091; 99% CI: 1.884–5.070, P–value < 0.001) were associated with CAs in the crude Odds ratio analysis and may be responsible for the occurrences of CAs. Differently, folic acid (COR = 0.563; 99% CI: 0.385–0.825, P–value < 0.001) was considered to have a protective effect against the occurrence or the development of CAs.

## Independent predictors (risk factors) of CAs adjusted for multiple tests

The variables with P- value of 0.2 and below in the bivariate analysis were entered into multivariable logistic regression model adjusted to observe exposure variables association with CAs (Tables 6 and 7). As a result, unidentified drug usage in the first three months of pregnancy (AOR = 3.435; 99% CI: 2. 012–5.863), exposure to pesticides (AOR = 3.926; 99% CI: 1.266–12.176), passive smoking (AOR = 4.104; 99% CI: 1.892–8.901), surface water as sources of drinking (AOR = 2.073; 99% CI: 1.221–3.519), no antenatal care visits (AOR = 2.952; 99% CI:1.166–7.472), 1 to 3 antenatal care visits (AOR = 2.121; 99% CI: 1.390–3.237) were significantly associated with the occurrence of CAs. On the other hand, iron folate / folic acid supplementation during the indexed pregnancy (AOR = 0.639; 99% CI: 0.247–0.740) had a protective effect againstthe development of CAs.

$$\log(odds\ of\ CA) = -1.997 + 1.234(drug\ use) + 1.368\ (pesticide\ exposure) + 1.412\ (passive\ smoker)$$
$$- 0.358\ (folic\ acid) + 1.349\ (history\ of\ CA + 0.729\ (surface\ water)$$
$$+ 1.082\ (no\ antenatal\ care\ visits) + 0.752\ (1\ to\ 3\ antenatal\ care\ visits)$$

**Table 6. Responses of study subjects on possible risk factors for CAs: Multivariable analysis of AOR for associated risk factors of CAs in southwestern Ethiopia.**

| Variables | | Cases | | Controls | | COR | AOR | 99% CI | |
|---|---|---|---|---|---|---|---|---|---|
| | | Number | % | Number | % | | | lower | upper |
| Drug use during the first three months of pregnancy | Yes | 57 | 22.7 | 77 | 8.7 | 3.076 | **3.435** | **2.012** | **5.863** |
| | No | 194 | 77.3 | 810 | 91.3 | | 1 | | |
| Exposure to pesticides | Yes | 13 | 5.2 | 12 | 1.4 | 3.998 | **3.926** | **1.266** | **12.176** |
| | No | 236 | 94.8 | 874 | 98.6 | | 1 | | |
| Passive smoking | Yes | 28 | 11.2 | 28 | 3.2 | 3.834 | **4.104** | **1.892** | **8.901** |
| | No | 223 | 88.8 | 859 | 96.8 | | 1 | | |
| Folic acid use | Yes | 89 | 35.5 | 438 | 49.4 | 0.568 | **0.639** | **0.247** | **0.740** |
| | No | 162 | 64.5 | 449 | 50.6 | | 1 | | |
| History of congenital anomalies | Yes | 7 | 2.8 | 7 | 0.8 | 2.303 | 3.741 | 0.875 | 16.006 |
| | No | 244 | 97.2 | 880 | 99.2 | | 1 | | |

CI: Confidence interval; COR: Crude Odds ratio; AOR: Adjusted Odds ratio.

**Table 7. Responses of study subjects on possible risk factors for CAs: Multivariable analysis of AOR for associated risk factors of CAs in southwestern Ethiopia.**

| Charactristics | Variables | Cases | | Controls | | AOR | 99% CI | |
|---|---|---|---|---|---|---|---|---|
| | | Number | % | Number | % | | Lower | Upper |
| The water source for drinking | Pipe water | 169 | 67.3 | 716 | 80.7 | 1 | | |
| | Underground water | 26 | 10.4 | 72 | 8.1 | 1.492 | 0.756 | 2.944 |
| | Surface water | 56 | 22.3 | 99 | 11.2 | **2.073** | **1.221** | **3.519** |
| Antenatal care follow up | No antenatal visits | 18 | 7.2 | 26 | 2.9 | **2.952** | **1.166** | **7.472** |
| | 1 to 3 visits | 138 | 55.2 | 310 | 35.1 | **2.121** | **1.390** | **3.237** |
| | Minimum of 4 visits | 94 | 37.6 | 548 | 62.0 | 1 | | |

CI: Confidence interval; AOR: Adjusted Odds ratio.

## Discussion

Several studies described that the human embryo is well protected in the uterus by the extra-embryonic membranes, although teratogens may cause developmental disruptions after maternal exposure to them in a specific period of organogenesis during the critical period in early pregnancy [10].

In our findings, maternal exposure to actual smoking in crude risk estimate, passive smoking during early pregnancy and exposure to pesticides and herbicides during the critical period of embryogenesis had a significant association with the occurrence of CAs. Moreover, case mothers (1.2%) who were smoking cigarettes during their pregnancy show a 6 times likely increased risk to have neonates with CAs as compared to the control mothers (0.2%).

In line with the present study, studies done in Iraq and Egypt show that maternal smoking either actual or passive smoking in the first three months of pregnancy was strongly associated with the occurrence of birth defects specifically cleft lip with or without cleft palate [14,15]. Congruent with our findings, a study done in southeastern Ethiopia, Bale zone, Oromia region by Mekonnen et al. [16] reported that maternal exposure to pesticides during early pregnancy had a strong association with the occurrence of CAs with the frequency of 13.2% in exposed compared to 4.2% in unexposed.

On the contrary, Taye et al [17] reported in their study done in Addis Ababa and Amhara region that cigarette smoking, either actual or passive smoking, was not associated with CAs. However, several studies, including the present study, showed that there is an association of smoking with the occurrence of CAs [18–22]. The difference might be due to the cultural difference in practicing cigarette smoking or staying with a smoker of a country with a geographically different location.

Likewise,Taye et al [17] reported that maternal exposure to chemicals during early pregnancy had a significant association with the occurrence of CAs. Unlike reports from the study done in Addis Ababa and Amhara region where alcohol had a strong association with the risk of having a child with CAs, in our findings, alcohol had no strong association with the occurrence of CAs. This difference might be linked to religious restriction as most of the participants in the present study were Muslims and Protestants who did not drink alcohol.

In our findings, unidentified medicinal use during early pregnancy had a strong association with the occurrence of CAs. Several findings also reported that the use of unidentified medicines during the early period of embryogenesis had a significant association with the occurrence of CAs [17,23,24].

According to our findings, the use of surface water as a source of drinking had a significant association with the occurrence CAs. This might be because chemicals and pesticides added to the surface water through several routes/sources can contaminate the surface water. Using

contaminated water as a source of drinking especially during early pregnancy may contribute to the occurrence of CAs as it introduces chemicals that may cross the placenta and disrupt embryonic differentiation during the period of embryogenesis especially in the first 8 weeks of the embryonic period. Our finding agrees with study that reported maternal exposure to drinking water containing nitrate had a risk of having a neonate with CAs [25].

Maternal diseases such as asthma and hypertension showed no association with the existence of CAs in the present study; Besides, maternal diabetes had no significantly associated with the risk of having a neonate with CAs. Incongruent to our findings, several reports from other scientific studies showed that maternal diabetes has been a well established risk factor for CAs and adverse birth outcomes [24,26]. Besides, several findings described that gestational diabetes has a strong association with fetal growth abnormalities. Furthermore, pre-gestational diabetes was identified to be an important risk factor for structural anomalies due to the terato-genic effect of poorly controlled diabetes and is considered to be the most important risk factor during early period of pregnancy, especial during the first 8 weeks, at which active differentia-tion of organ systems could occur [24,27].

Correa (2016) [28] reported that the embryopathy associated with pre-gestational diabetes mellitus is nonspecific underlying metabolic disorders disturbing morhogenetic process. Accordingly, maternal hyperglycemia results in increased glucose levels in the embryo. Conse-quently, biochemical aberrations increase oxidative stress that leads to cellular apoptosis.

Oxidative stress results in inhibition of the Pax3 gene especially in the processes of neurula-tion [28]. Besides, oxidative stress occurred due to the imbalance between the production of oxygen free radicals and the antioxidant defense mechanism of the cells which can generate the irreversible oxidation of DNA leading to apoptosis as a result of enzymatic inactivation [29,30].

The present study showed that there was a strong association between insufficient maternal folic acid supplementation during early pregnancy and the occurrence of CAs. Mothers of the newborn who did get folic acid supplementation during their early pregnancy were strongly protected against having neonates with CAs, specifically neural tube defects. In other words, mothers of the newborns who did not get folic acid supplementation during their early preg-nancy were twice as likely to have a baby with CAs. Several studies reported that folic acid sup-plementation in early pregnancy reduces the occurrence of specific CAs.

In this study, mother of the cases who did not get folic acid supplementation during early pregnancy were 64.5%, revealing that there is poor folic acid supplementation in southwestern Ethiopia. In agreement with our findings, poor folic acid supplementation also observed in the Amhara region, northern Ethiopia [23]. However, a study in the Tigray region, northeastern Ethiopia, revealed that 40.9% participants were adhered to iron folate supplementation [31].

Although, the Federal Ministry of Health and the Regional Health Bureau both promote folic acid supplementation to all women during their indexed pregnancy, most pregnant women have not been provided such supplements yet. This might be because of poor quality of antenatal care in the study region and in Ethiopia in general or communities' resistance because of lack of knowledge about the importance of folic acid supplementation.

According to the present study, 7.2% of mothers of the cases and 2.9% of mothers of the controls received no antenatal care follow up. The difference between the cases and the con-trols was statistically significant indicating that the mothers who had not received antenatal care during their pregnancy were strongly associated with having a baby with CAs. Similarly, mothers who had received 1 to 3 antenatal care visits had a significant association with the presence of CAs compared to those who had four and above antenatal care visits.

In our findings, although maternal illness and the use of antibiotics during their pregnancy showed association with the occurrence of CAs, it is not significant statistically. Incongruent with this study, several studies showed that maternal illness and the use of antibiotics during

early pregnancy has a strong association with the occurrence of CAs [17,23]. The inconsistency between the present study and other reports might be due to participants fail to report the actual chronic illness that they had during pregnancy.

Although exposure to radiation, especially X–rays, during early pregnancy seems to have an association with the occurrence of CAs in its crude risk estimate, multivariable logistic regression shows no association with the occurrence of the CAs.

In our findings, a maternal history of abortion and stillbirth has no association with the occurrence of CAs. Besides, the mother's previous birth history of CAs and the birth history of CAs in the family showed no association with the occurrence of CAs. This may have genetic implications. In our findings, more live births were observed in controls than in cases. On the contrary, more stillbirths (61.6%) were observed in cases of newborns with CAs, as compared to the controls (5.2%). A similar report showed that the overall CA-specific stillbirth risk was increased among affected fetuses over the occurrence of stillbirth in the general population in the United State [32,33].

In the present study, about 39.9% of the neonates born with CAs had low birth weight revealing that the presence of CAs can affect the birth weight of the neonate contributing to fetal growth restriction. A study done in northern Ethiopia indicated that parity, low birth weight, gestational age less than 35 weeks, male sex, and lack of antenatal care were significantly associated with CAs [34].

Another study indicated that the prevalence rates of low birth weight and premature birth were significantly greater among infants with CAs than in their non-afflicted counterparts [35]. In agreement with previous studies, the result of the present study indicated that 47.6% of the neonates with CAs were preterm (premature) births where a statistically significant difference was observed between the cases and controls, indicating that the occurrence of CAs contributes to premature delivery with low birth weight.

In this study, socio-demographic characteristics such as maternal educational level, average monthly income, maternal occupation, and paternal age showed no significant association with the occurrence of CAs. Similarly, parity, gravidity, the onset of labor, mode of delivery, and types of birth outcome and birth order of the newborns showed no significant association with the occurrence of CAs.

## Strength of the study

First, we used hospitals with high caseloads in southwestern Ethiopia where most cases of CAs were expected. Secondly, a case-control study design was used with a maximum case to control ratio of 1:4. We used trained health professional data collectors, including gynecologists who evaluated the presence of CAs (the cases). The controls were randomly selected from the neonates without CAs whereby two of them selected from newborns delivered 24 hours before the occurrence of the case and the other two were randomly selected from newborns delivered immediately after the occurrence of the case. This might increase the efficiency of the study result in providing predictors or associated factors. Finally, the data analysis was based on all maternal, neonatal, and other associated factors that may contribute to the occurrence of CAs.

## Limitation of the study

From existing hospitals in the study regions, only six selected hospitals were used for the study, although cases of CAs were expected in the remaining hospitals and health stations, which may limit generalizability to the wider population of Ethiopia. Secondly, it was a hospital-based study design and might have missed cases of CAs for deliveries that occurred outside the

study hospitals within the community. Thirdly, although, structured data collection tools were used, participants recall and self–report of some factors might introduce bias.

## Conclusions

In the findings of the present study, maternal socio-demographic factors such as maternal educational level, average monthly income, maternal occupation, and paternal age showed no significant association with the occurrence of CAs. Associated risk factors such as maternal active or passive smoking, exposure to pesticides, exposure to chemicals, use of unidentified medicine during the first three months, use of surface water for drinking had a significant association with the occurrence of CAs. Differently, the use of folic acid during the indexed pregnancy had a significant protective effect against the occurrence of CAs. However, poor folic acid supplementations were observed in the country, Ethiopia. There is a need to continuously provide health information for the community on how to prevent the occurrence of CAs and there is a need to improve the quality of antenatal care follow up as well as folic acid supplementation mainly through food fortifications.

## Supporting information

**S1 Table. Descriptive analysis (mean and Std.) Deviation of maternal and neonatal characteristics of the study participants in relation to birth weight, gestational age, birth order of the infancy, maternal age, paternal age, maternal average monthly income, party and gravida.**
(DOCX)

**S2 Table. Interdependent risk factors adjusted for multiple tests.**
(DOCX)

**S1 Annexes.**
(DOCX)

**S1 File.**
(SAV)

**S1 Syntax.**
(SPS)

## Acknowledgments

We would like to thank all medical wards of each study hospital in southwestern Ethiopia who helped us in providing the information necessary for the study. We would like to extend our sincere thanks to Jimma University specialized hospital, Shanene Gibe hospital, Limu Genet hospital, Mettu Karl hospital, and Nekemte hospital in southwestern Ethiopia, who allowed and helped us during data collection at their centers. Our gratitude extends to our trained data collectors at each hospital of our study area. Our thanks also extend to the staff of the departments of gynecology and obstetrics of each hospital who had direct or indirect inputs for the successful accomplishment of the study.

## Author Contributions

**Conceptualization:** Soressa Abebe.

**Data curation:** Soressa Abebe.

**Formal analysis:** Soressa Abebe, Girmai Gebru, Demisew Amenu, Zeleke Mekonnen, Lemessa Dube.

**Funding acquisition:** Soressa Abebe.

**Investigation:** Soressa Abebe, Girmai Gebru, Demisew Amenu, Zeleke Mekonnen, Lemessa Dube.

**Methodology:** Soressa Abebe, Girmai Gebru, Demisew Amenu, Zeleke Mekonnen, Lemessa Dube.

**Project administration:** Soressa Abebe.

**Resources:** Soressa Abebe.

**Software:** Soressa Abebe.

**Supervision:** Soressa Abebe, Girmai Gebru, Demisew Amenu, Zeleke Mekonnen, Lemessa Dube.

**Validation:** Soressa Abebe, Girmai Gebru, Demisew Amenu.

**Visualization:** Soressa Abebe, Zeleke Mekonnen, Lemessa Dube.

**Writing – original draft:** Soressa Abebe.

**Writing – review & editing:** Soressa Abebe.

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
