## [Decision Letter · Decision Letter 0]

7 Oct 2020

PONE-D-20-23170

Risk Factors Associated with Congenital Anomalies among Newborns in Southwestern Ethiopia: A case-control study

PLOS ONE

Dear Dr. Geneti,

Thank you for submitting your manuscript to PLOS ONE. After careful consideration, we feel that it has merit but does not fully meet PLOS ONE’s publication criteria as it currently stands. Therefore, we invite you to submit a revised version of the manuscript that addresses the points raised during the review process.

We look forward to receiving your revised manuscript.

Kind regards,

Linglin Xie

Academic Editor

PLOS ONE

Journal Requirements:

2. Please include additional information regarding the checklist and questionnaire used in the study and ensure that you have provided sufficient details that others could replicate the analyses.

For instance, if you developed a questionnaire and/or checklist as part of this study and it is not under a copyright more restrictive than CC-BY, please include a copy, in both the original language and English, as Supporting Information.

'The authors, Mr. Soressa Abebe and Dr. Girmai Gebru receive salary from Addis Ababa University, Dr. Demisew Amanu, Professor Zeleke Mekonnen and Mr. Lemessa Dube receive salary from Jimma University.

The authors have declared that no competing interests exist.'

5. Please amend your list of authors on the manuscript to ensure that each author is linked to an affiliation. Authors’ affiliations should reflect the institution where the work was done (if authors moved subsequently, you can also list the new affiliation stating “current affiliation:….” as necessary).

Reviewers' comments:

Reviewer's Responses to Questions

**Comments to the Author**

1. Is the manuscript technically sound, and do the data support the conclusions?

Reviewer #1: Partly

2. Has the statistical analysis been performed appropriately and rigorously? 

Reviewer #1: No

3. Have the authors made all data underlying the findings in their manuscript fully available?

Reviewer #1: No

4. Is the manuscript presented in an intelligible fashion and written in standard English?

Reviewer #1: Yes

5. Review Comments to the Author

Reviewer #1: Major comments:

This article is a retrospective study on the association between maternal exposure to risk factors during early pregnancy and the development of congenital anomalies. The study was conducted based on data collected from six selected hospitals in southwestern Ethiopia through two years. The data analysis provided exciting data and analysis on the possible correlation between possible risk factors and congenital anomalies.

One issue with the experimental design is the sampling method to select the six hospitals. The authors mentioned that the hospitals were selected purposively based on their capacity to carry out the required assessment. This non-randomness of sampling may sharply limit the generalization of the conclusion to newborns in the whole area. This point has been mentioned in the “Limitation of the Study” section. But without providing more evidence that these hospitals can represent all the hospitals in the area, the authors should be more careful when they tried to generalize the conclusions.

One major issue with the statistical analyses is that the author conducted multiple tests during their studies, while they used a threshold P-value of 0.05 for significance, which is a typical cut-off when doing a single test. I would suggest that they use an adjusted p-value instead to control the false positive rate for this multiple test case.

Also, it would be more informative if a formula for the final regression model with corresponding coefficients was provided beside the descriptions.

Also, there should be a separate section where data availability is described. The authors should show where the original questionnaire and raw data can be found for their readers.

Minor comments:

Table 7: It would be clearer if the variable name is in a separate column from the variable levels.

6. PLOS authors have the option to publish the peer review history of their article (what does this mean?). If published, this will include your full peer review and any attached files.

Reviewer #1: No

---

## [Author Response · Author response to Decision Letter 0]

10 Dec 2020

• A rebuttal letter that responds to each point raised by the academic editor and reviewer(s). You should upload this letter as a separate file labeled ‘Response to Reviewers.’

Comment accepted. The rebuttal letter that responds to each point raised by the academic editor and reviewer(s) is included. The letter is uploaded as a separate file and labeled ‘Response to Reviewers’

• A marked-up copy of your manuscript that highlights changes made to the original version. You should upload this as a separate file labeled ‘Revised Manuscript with Track Changes’

Comment accepted. The changes made to the original version is highlighted in orange color in the revised manuscript. The file is labeled as Revised Manuscript with Track Changes’.

• An unmarked version of your revised paper without tracked changes. You should upload this as a separate file labeled ’‘Manuscript'.

Comment accepted. The unmarked version of the revised paper without tracked changes labeled as ‘Manuscript’

• The financial disclosure and updated statement included in the cover letter

Journal Requirements:When submitting your revision, we need you to address these additional requirements.

1. Please ensure that your manuscript meets PLOS ONE’s style requirements, including those for file naming. 

 Dear editor,

We thank you for reminding us to check our manuscript in lines with PLOS ONE's style requirements. We checked journal formats and corrected it accordingly.

2. Please include additional information regarding the checklist and questionnaire used in the study and ensure that you have provided sufficient details that others could replicate the analyses.

For instance, if you developed a questionnaire and/or checklist as part of this study and it is not under copyright more restrictive than CC-BY, please include a copy, in both the original language and English, as Supporting Information.

 Dear editor,

 We uploaded the questionnaire written in English, Afaan Oromo, and Amharic languages as supporting information.

We confirm this does not alter our adherence to PLOS ONE policies on sharing data and materials.

Funding

This study obtained findings s from Addis Ababa University and Jimma University for data collection. The funding universities do not have any role in the study's design, data collection, analysis and interpretation of data and writing of the manuscript. 

We included the updated competing interest statement in our cover letter.

The corresponding author declared on behalf of all authors

The corresponding author linked his ORCID iD

We included the updated list of authors and their corresponding affiliation as per request.

Response to Reviewers' comments:

 We thank you for your invaluable comments, suggestions and recommendations. We have amended the manuscript based on your constructive comments as follows:

 We thank you for raising this critical comment; although we mentioned purposive selection of the hospitals, we have included all hospitals except one. So, we feel it can be generalizable to the whole study area. Besides our indication of the limitations, we have now updated the revised manuscript indicating that almost all hospitals six out of seven in the study area were assessed. 

Thank you again for your constructive comment. We just considered P-value for candidate selection. We used the 95% CI interval for the final decision to minimize complications/false positive rate. If the confidence interval includes a null value, we declared not statistically significant. 

 Thank you for the comment. We Now included the final regression formula as per your invaluable recommendation as follows.

log⁡(odds of CA)=-1.997+1.234(drug use)+1.368 (pesticide exposure)+1.412 (passive smoker)-0.358 (folic acid)+1.349 (history of CA+0.729 (surface water)+1.082 (no antenatal care visits)+0.752 (1 to 3 antenatal care visit).

Thanks for the suggestion about the availability of data and materials. 

We uploaded all required documents as supporting information including consent form and questionnaire written in the English, Afaan Oromo, and Amharic languages. 

As per your recommendation and PLOS ONE standard of minimal data set, particularly in our manuscript, we a

Availability of data and materials

All relevant data are within the manuscript and its supporting information. 

We have corrected.

---

## [Decision Letter · Decision Letter 1]

16 Dec 2020

PONE-D-20-23170R1

Risk factors associated with congenital anomalies among newborns in southwestern Ethiopia: A case-control study

PLOS ONE

Dear Dr. Geneti,

Thank you for submitting your manuscript to PLOS ONE. After careful consideration, we feel that it has merit but does not fully meet PLOS ONE’s publication criteria as it currently stands. Therefore, we invite you to submit a revised version of the manuscript that addresses the points raised during the review process.

We look forward to receiving your revised manuscript.

Kind regards,

Linglin Xie

Academic Editor

PLOS ONE

Reviewers' comments:

Reviewer's Responses to Questions

**Comments to the Author**

1. If the authors have adequately addressed your comments raised in a previous round of review and you feel that this manuscript is now acceptable for publication, you may indicate that here to bypass the “Comments to the Author” section, enter your conflict of interest statement in the “Confidential to Editor” section, and submit your "Accept" recommendation.

Reviewer #1: All comments have been addressed

2. Is the manuscript technically sound, and do the data support the conclusions?

Reviewer #1: Yes

3. Has the statistical analysis been performed appropriately and rigorously? 

Reviewer #1: N/A

4. Have the authors made all data underlying the findings in their manuscript fully available?

Reviewer #1: Yes

5. Is the manuscript presented in an intelligible fashion and written in standard English?

Reviewer #1: Yes

6. Review Comments to the Author

Reviewer #1: The modifications are reasonable to all comments except the one for the multiple test issue. The multiple test issue would not be solved by replacing p-values with significance intervals. If a 95% CI is used, then significance level is still 5%. The authors should also adjust their significance levels and choose a more restrict CI to adjust for the multiple test issue.

7. PLOS authors have the option to publish the peer review history of their article (what does this mean?). If published, this will include your full peer review and any attached files.

Reviewer #1: No

---

## [Author Response · Author response to Decision Letter 1]

7 Jan 2021

Dear editor,

We thank you for your invaluable comments, suggestions and recommendations in our revised manuscript,

Now we included responses and corrections as per your comments and suggestions for each point raised either by the editor or the reviewer’s as follows

The rebuttal letter is prepared and uploaded as a separate file and labeled ‘Response to Reviewers’

• A marked-up copy of your manuscript that highlights changes made to the original version. You should upload this as a separate file labeled 'Revised Manuscript with Track Changes'

The changes made to the original version are highlighted in orange color in the revised manuscript. The file is labeled as ‘Revised Manuscript with Track Changes’.

The unmarked version of the revised paper without tracked changes labeled as ‘Manuscript’

• � No changes made

Reviewers' comments:

Reviewer's Responses to Questions

Comments to the Author

1. If the authors have adequately addressed your comments raised in a previous round of review and you feel that this manuscript is now acceptable for publication, you may indicate that here to bypass the “Comments to the Author” section, enter your conflict of interest statement in the “Confidential to Editor” section, and submit your "Accept" recommendation.

Reviewer #1: All comments have been addressed

2. Is the manuscript technically sound, and do the data support the conclusions?

Reviewer #1: Yes

3. Has the statistical analysis been performed appropriately and rigorously?

 Reviewer #1: N/A

4. Have the authors made all data underlying the findings in their manuscript fully available?

Reviewer #1: Yes

5. Is the manuscript presented in an intelligible fashion and written in Standard English?

Reviewer #1: Yes

6. Review Comments to the Author

Reviewer #1: The modifications are reasonable to all comments except the one for the multiple test issue. The multiple test issue would not be solved by replacing p-values with significance intervals. If a 95% CI is used, then significance level is still 5%. The authors should also adjust their significance levels and choose a more restrict CI to adjust for the multiple test issue.

Comment accepted: 

Dear reviewer, we thank you for raising this critical comment. It is acceptable that instead of looking at the 95% CI, logistic regression analyses was repeated by using a narrower interval (e.g. 99% CI) to reduce the significance level from 0.05 to 0.01. Setting a higher significance threshold for individual comparisons to compensate for the number of inferences was being made. This is one way that we can reduce the probability of getting a false positive. Therefore, as per your invaluable comment, we corrected and adjusted the multiple tests to 99% confidence interval and significance level of 0.01(1%) and indicated it in the revised manuscript in (table 6 and 7) as well as in the text under independent predicator (risk factors) of congenital anomalies.

Interdependent risk factors adjusted for multiple tests are indicated and uploaded as supporting information as S2 Table for your considerations.

Best regards!

---

## [Editor Report · Decision Letter 2]

11 Jan 2021

Risk factors associated with congenital anomalies among newborns in southwestern Ethiopia: A case-control study

PONE-D-20-23170R2

Dear Dr. Geneti,

We’re pleased to inform you that your manuscript has been judged scientifically suitable for publication and will be formally accepted for publication once it meets all outstanding technical requirements.

Kind regards,

Linglin Xie

Academic Editor

PLOS ONE
---

## [Editor Report · Acceptance letter]

18 Jan 2021

PONE-D-20-23170R2 

Risk factors associated with congenital anomalies among newborns in southwestern Ethiopia: A case-control study 

Dear Dr. Abebe:

I'm pleased to inform you that your manuscript has been deemed suitable for publication in PLOS ONE. Congratulations! Your manuscript is now with our production department. 

Kind regards, 

on behalf of

Dr. Linglin Xie 

Academic Editor

PLOS ONE